# Data Processing of SPR Curve Data to Maximize the Extraction of Changes in Electrochemical SPR Measurements

**DOI:** 10.3390/bios12080615

**Published:** 2022-08-08

**Authors:** Suzuyo Inoue, Kenta Fukada, Katsuyoshi Hayashi, Michiko Seyama

**Affiliations:** NTT Device Technology Laboratory, NTT Corporation, Atsugi 243-0198, Kanagawa, Japan

**Keywords:** surface plasmon resonance (SPR), electrochemical SPR (EC-SPR), signal processing, glutamate

## Abstract

We developed a novel measuring and data-processing method for performing electrochemical surface plasmon resonance (EC-SPR) on sensor surfaces for which detecting a specific SPR angle is difficult, such as a polymer having a non-uniform thickness with coloration. SPR measurements are used in medicine and basic research as an analytical method capable of molecular detection without labeling. However, SPR is not good for detecting small molecules with small refractive index changes. The proposed EC-SPR, which combines SPR measurements with an electrochemical reaction, makes it possible to measure small molecules without increasing the number of measurement steps. A drawback of EC-SPR is that it is difficult to detect a specific SPR angle on electron mediators, and it was found that it may not be possible to capture all the features produced. The novel method we describe here is different from the conventional one in which a specific SPR angle is obtained from an SPR curve; rather, it processes the SPR curve itself and can efficiently aggregate the feature displacements in the SPR curves that are dispersed through multiple angles. As an application, we used our method to detect small concentrations of H_2_O_2_ (LOD 0.7 μM) and glutamate (LOD 5 μM).

## 1. Introduction

Surface plasmon resonance (SPR) is a versatile measurement method that has a relatively simple optical configuration and can measure t refractive index changes with high sensitivity (10^−5^ to 10^−6^) near a gold surface (<200 nm) in real-time. The target molecule in a sample solution can be measured without sensitization, such as fluorescent labeling by immobilizing a molecule-specific modification, for example, an antibody, on a gold thin film within the observation region [1]. SPR is typically used in basic biochemistry research [1], drug discovery development [2], and food safety [3]. Non-label SPR measurements are also useful as a point of care testing (POCT) technique [4], where they are advantageous for reducing the number of measurement steps. We developed a 1D-surface plasmon resonance (1D-SPR) instrument and a passive microfluidic chip [5] and reported the simultaneous measurement of multiple molecules using arrayed antibodies [6,7] and DNA aptamer for POCT [8] and the measurement of the near-wall molecular transport rate in a microchannel [9].

However, the SPR measurements have the general problem of lowering or loss of sensitivity when the sample substance is absorptive to the light used in the measurement. The light wavelength of the SPR measurement must be carefully selected in order to avoid absorption by samples including water, proteins, amino acids, and so on, but not all cases can be covered. One such situation is an electrochemical SPR (EC-SPR) measurement that combines an electrochemical reaction with an SPR measurement [10].

Figure 1 is a schematic illustration of our EC-SPR measurement. A measurement chip in which an electron mediator, an oxidizing enzyme of a target molecule, and a hydrogen peroxide protein ((horseradish peroxidase (HRP)) are immobilized is prepared on a gold thin film, and a sample solution is reacted on it. When the target molecule reacts with the oxidizing enzyme, the electron transfer shown in the figure occurs, ultimately changing the charge of the electron mediator. As the charge of the electron mediator changes, the number of charge compensations present around the electron mediator changes, resulting in a refractive index change. In this case, charge compensation is considered to be dominated by chloride and phosphate ions contained in the buffer. This molecular migration causes a refractive index change, and the electrochemical reaction is converted into an SPR signal. Since many small molecules, such as amino acids and sugars, are present in the molecules to be measured [11,12,13,14], it is desirable to be able to measure small molecules even though the change in refractive index due to them is not large. To compensate for the small refractive index change, researchers previously used sandwich assays in which secondary antibodies are reacted after target-molecule capture [15,16,17]. These approaches are excellent for sensitizing target capture sensitivity and are suitable for laboratory use. However, these countermeasures eliminate the benefits of SPR, which reduces the number of measurement steps for POCT. A simple method for measuring small molecules without increasing the number of measuring steps is required. Here, EC-SPR, which utilizes electrochemical sensitization to detect small molecules, is a promising way to overcome the weakness, as many have reported [18,19].

However, drawbacks with EC-SPR measurements still remain. When the electrodes are inserted from the outside of the measurement chip, errors occur in the positioning of the electrodes from one measurement to the next, and passive liquid feed handling using capillary force becomes difficult. Therefore, it is necessary to modify the electron mediator on the planar electrode to transfer electrons to the working electrode. The electron mediator layer changes its oxidation state to donate/accept electrons, which results in a change in the optical absorption and/or its thickness. Determination of the refractive index based on the minimum point of the SPR curve, which is correlated with the incident angle of light (SPR angle), is widely used. For two-dimensional SPR measurements, the actual detected parameter is light intensity, not SPR angle; the basic concept is that the light intensity indicates the point on the SPR curve that correlates with the SPR angle. However, when light is absorbed, the SPR curve becomes different from its ideal form, and this causes an error in the determination of the minimum point, making it difficult to calculate an accurate SPR angle.

In this work, we first performed cyclic voltammetry (CV) measurements on an EC-SPR electrode using a multi-analysis 1D-SPR measurement system. Our 1D-SPR sensor generates a data set in which the X coordinate corresponds to the SPR angle, and the Y coordinate corresponds to the microfluidic channel position together with the intensity at each (X, Y) cell in every measurement time period (normally 1 s), so the structure of the data is similar to an image. As a result, we found that on an EC-SPR electrode, changes in the state transition of the electron mediator do not only appear with changes in one incident angle (SPR angle) but also with variations in the vector due to multiple incident angles and that it is difficult to form a small mediator layer uniformly. Thus, we propose a new concept to extract the refractive index changes of the target layer without calculating a specific SPR angle to find a minimum point on the SPR curve. This method is based on post-data processing, with Karhunen–Loeve (KL) conversion [20], of an SPR-acquired image.

KL conversion is similar in concept to principal component analysis (PCA) which is used to extract the features of two-dimensional image data with the average values of the target image. Recently, the PCA data-processing technique has been employed for biosensing data analyses to obtain vector quantization [21,22,23,24], and its high feature extraction ability has been demonstrated. Therefore, a similar effect is expected even if KL conversion is based on the same principle. The major difference between KL conversion and PCA is that, in contrast to PCA, which generally decomposes the covariance matrix of data into eigenvalues, KL conversion decomposes the correlation function (covariance normalized by standard deviation) of data into eigenvalues to obtain feature quantities. In other words, KL conversion is a data analysis method that requires the measurement of the maximum and minimum change state of the object to be measured as a prerequisite for the data to be applied [25]. When this is considered in place of the measurement protocol, the error component of the electrode surface, depending on the actual measurement to be performed, is acquired every time. This makes KL conversion an optimum method for analyzing electrochemical measurement in non-uniform mediator application conditions. In the electrochemical SPR measurement, the maximum state change corresponds to the complete oxidation state of the mediator, and the minimum state change corresponds to the complete reduction state of the mediator. Since these conditions can be easily set by applying voltage with a potentiostat, the measurement protocol can be adjusted simply by customizing it according to the acquisition of data necessary for analysis. As described above, the application of KL conversion to EC-SPR data analysis enables the simultaneous processing of both the ideal response according to the physical law of the measurement principle and the non-ideal response depending on the actual measurement system without a large measurement effort.

Here, we report a novel post-data processing method for EC-SPR measurements using multi-analysis 1D-SPR instruments, a measurement chip and a protocol for its friendly data collection. As an application, we detected H_2_O_2_ and glutamate from small sample solutions of 10 μL.

## 2. Materials and Methods

### 2.1. Fabrication of EC-SPR Measurement Chip

Figure 2A show a schematic representation and a photograph of the SPR sensor chip. We used BK7 glass for the bottom substrate of the measurement chip and acryl for the top substrate. The BK7 glass substrates were first cleaned by washing them with neutral detergent (Clean Ace S, As One Corp., Osaka, Japan). They were then washed with deionized water and dried under a nitrogen flow. The acryl substrates were cut using a laser cutter (VLS2.30, Universal Laser Systems, Inc., Scottsdale, AZ, USA), and the electrode hole was set diagonally so that the back electrode could be contacted. The cut acryl substrates were washed with acetone, ethanol, and deionized water (D.W.) in sequence, followed by drying under a nitrogen flow. Next, 5-nm-thick titanium and 45-nm-thick gold thin films were sputtered on the BK7 glass substrate and acryl substrate by sputtering equipment (QAM-4, ULVAC, Chigasaki, Japan). The gold film was patterned with the stencil method using dicing tape (Elegrip tape, Denka corp., Tomakomai, Japan). 

To develop the reference electrode, we applied an Ag/AgCl ink (Ag/AgCl ink for the reference electrode, BAS, Tokyo, Japan) with D.W. to the BK glass substrate. To develop the working electrode, we treated the sensing surface with two layers, i.e., an electron mediator and HRP layer and the target oxidase layer. First, we stamped 10-fold diluted osmium polymer with HRP (osmium polymer, BAS, Tokyo, Japan) using a 1-mm-diameter PDMS stamp cut out by a biopsy trepan (KAI Corporation, Tokyo, Japan) to treat the electron mediator and HRP layer, as shown in Figure 2B. The pattern of the current electrode was developed in the acryl substrate. Dissolution of the gold thin film was prevented through electrolysis in the vertical direction from the ceiling to the bottom of the microchannel instead of performing it in the direction of the flow channel by arranging the three electrodes on one substrate. For glutamate oxidase immobilization on the electron mediator polymer, we placed 0.2 μL of 1 wt% Poly-L lysine (Sigma-Aldrich, St. Louis, MO, USA) in deionized water, 0.2 μL of 3 unit/mL GluOX (Sigma-Aldrich, St. Louis, MO, USA) in 50 mM potassium phosphate buffer (pH 7.4), and 0.2 μL of 1 wt% gellan gum (FUJIFILM Wako Pure Chemical Corp., Osaka, Japan) in deionized water in sequence onto the working electrode in the BK7 glass substrate using a micropipette.

The BK7 glass substrate and acryl substrate were attached with double-sided tape in a vise. A 75-μm-high and 1-mm-wide microchannel was fabricated between the substrates, and 3 mm inlets and 1 mm outlets were also fabricated. To prevent leakage of the solution, we placed a PDMS block between the outlet and the pump tube. Short lead wires were soldered to the end of each electrode to provide conduction with the potentiostat.

### 2.2. Sample Flow Handling System

As a flow handling method, we used the “stop and flow system” that we previously developed [9]. This flow system enables us to stop the flow automatically once the inlet is empty and start the flow when another sample is injected into the inlet by controlling the power balance between the capillary force generated at the entrance of the flow channel and the constant negative pressure generated by an external pump. This method has the advantage of ease of handling because the traction force of the negative pump is constant throughout. Our flow system has a constant negative pump pressure from 0.5 to 7.6 mbar, as calculated from the microchannel aspect, the contact angle of the substrate, and sample solution viscosity. We set the negative pump pressure for operating to 7 mbar so that we could quickly change the sample solutions in the flow channel.

### 2.3. EC-SPR Measurement

The EC-SPR measurement chip was placed on the prism of the 1-D SPR instrument in refractive index preparation oil (certified refractive index oil, Cargille Laboratories, Cedar Grove, NJ, USA). It was connected to the potentiostat (Ivium CompactStat.h, Hokuto Denko Corp., Tokyo, Japan) via lead wires attached to the pads and connected to the pump (MFCS, Fluigent, Le Kremlin-Bicêtre, France) via a polyurethane tube in order to apply constant negative pressure to the fluidic channel.

Before the SPR measurement, the microchannel was filled with PBS and allowed to stand for 15 min to moisten the polymers on the working electrode. After moistening, we began the SPR measurement and applied voltages of 0.5 V and 0 V to the measurement chip for 20 s each to equalize the charge of the electron mediator and obtain data on changes in the SPR completely oxidized and reduced states. After measuring the states, the applied voltage was changed to 0.2 V, and 10 μL of the sample solution were injected into the chip inlet. Once the sample solution filled the microchannel, and the flow stopped automatically, the charge sweep was stopped, and the potentiostat was set to a current regulation mode of a current of 3 nA. In this state, the current from the electrode oxidizes the osmium ions at a constant rate, and the electrode potential increases accordingly. To prevent the value from exceeding the 0.5-V-applied measurement value set as the maximum change amount, the direction of the current is changed when this potential exceeds 0.3 V.

To measure the H_2_O_2_ concentration, H_2_O_2_ was diluted with PBS and solutions with concentrations ranging from 0 μM to 196 μM were prepared. To measure the glutamate concentration, glutamate was diluted with PBS and solutions ranging from 0 μM to 100 μM were prepared.

### 2.4. Data Analysis of EC-SPR Measurement

We used a portable 1-D SPR measurement instrument (SmartSPR, NTT Advanced Technology, Tokyo, Japan) and in-house software written using LabView 8.2 (National Instruments, Austin, TX, USA) to collect and process the data. The raw output data were image data containing measurement-position-dependent SPR curve information per time frame. The detailed data processing, for example, KL conversion, was performed using MATLAB software (MathWorks, Natick, MA, USA). The observation position showing the largest change in the SPR curve in the range where the electron mediator was applied used for analysis (Appendix A).

## 3. Results

### 3.1. Standard SPR Measurement at the Electron Mediator Polymer-Treated Electrode

Here, we examine the problem of the angle reading (standard) SPR measurement with the electron mediator polymer-treated electrode. Our 1-D SPR instrument obtains SPR curve data by collecting time-course image data, and we performed SPR measurements while sweeping the charge applied to the working electrode surface as cyclic voltammetry (CV) measurement (Figure 3A). Normally, when using gold film with an unmodified surface or a surface modified with a non-colorable material, the obtained SPR curve (Amount of light relative to the angle of incidence) has a large reduction in light intensity at a particular angle of incidence called the “SPR angle”. Moreover, the change in the SPR angle is related to the refractive index change near the gold surface. However, the osmium-based electron mediator polymer has uneven light absorption that deforms the SPR curve. Figure 3B shows the SPR curve of the gold thin film coated with osmium polymer in a completely oxidized state (0.5 V applied), completely reduced state (0 V applied), an intermediate state (0.25 V applied) during CV measurement. As described in the explanation of the principle, the EC-SPR measures the change in the number of counter ions due to the redox state of the electron mediator. The SPR measurement obtained on the electrode where the CV measurement is performed measures the state transition reaction (redox reaction rate) of the electron mediator to the electrode reaction and the change in the concentration of the electron mediator in each state. Immediately after the start of the sweep (0.5 V), the rate of change of the electron mediator from the oxidized to the reduced type is sufficiently low. Although the rate of change from the reduced to the oxidized type is large, there is almost no amount of the reduced type of electron mediator. Therefore, the state transition reaction of osmium proceeds slowly, but the transition reaction gradually accelerates as the electrode reaction by the sweep increases. When the scanning is carried out in the vicinity of 0.25 V, which is the peak potential of osmium, the state transition reaction of the majority of the osmium ends, so the state transition reaction rapidly decreases and gradually converges at 0 V. Therefore, the SPR measurement on the electrode should show a gradual change from 0.5 V to nearly 0.25 V, a large change at around 0.25 V, and a gradual change again at 0 V. Figure 3A shows the results of taking a broad view of the light intensity variation of the entire SPR curve. This seems to show the state change described above, but it is difficult to calculate the effect of changing the voltage on the SPR curve at the angle of the greatest reduction in light intensity at each time frame. In the case of osmium polymer, the light intensity may be reduced at certain angles or over a range of incident angles, such as the secondary dimming angle and at incident angles greater than 70 degrees (Figure 3B). In addition, when we focused on the vector of the light intensity change with respect to the voltage sweep at the incident light angle where the decrease in the light intensity can be confirmed, it was found that positive and negative coexist. Specifically, when the potential was swept from 0.5 V to 0 V, the light intensity of the first dimming remained almost unchanged, but that of the second dimming increased overall with a shift of the minimum intensity angle to the right, and the light intensity at angles greater than 70 degrees decreased overall. According to previous research [26], the real part of the refractive index contains information on the composition of the material, and the imaginary part of the refractive index is the light absorbed by the film on the gold and is revealed in the formation of the thickness and layer structure of the material. If only one component changes, the curve will only translate and retain its shape. However, when both components change, the linear relationship between the incident angle and the refractive index is broken, and the curve’s shape becomes distorted. As shown in Figure 4, the polymer’s thickness was not uniform; the osmium polymer treatment was performed by stamping for ease of manufacture, making it hard to make reproducibly uniform films as would be needed to ensure the accuracy of the refractive index. Moreover, the polymer was affected by the so-called coffee-ring effect and the solute, osmium and HRP moved to the edge of the stamp area as it dried. Under such conditions, it would be hard to detect the effect of the voltage applied to the working electrode only by applying noise processing to help detect one SPR angle.

### 3.2. Data Processing Using KL Conversion

As mentioned in Section 3.1, when the electron mediator is applied in the SPR observation area, the change in the SPR curve cannot be defined as a change in one SPR angle and is dispersed to various incident angles. Therefore, it is necessary to define the amount of one change by integrating these pieces of information. For this reason, we used all of the SPR curves, i.e., whole SPR image data, not just a single point of SPR angle data. It is necessary to select an appropriate data processing method. In the case of the EC-SPR measurement, the maximum voltage corresponds to the completely oxidized state, i.e., and the minimum voltage corresponds to the completely reduced state, as defined by the electron mediator used. To perform quantitative measurements, it is desirable that the change between the maximum voltage and the minimum voltage be linear. For that reason, we decided to perform KL conversion on the SPR image data, not to use normal PCA analysis.

KL conversion is a linear conversion method that is used for image compression. It examines the statistical information of the input data and converts a signal vector with the correlation between its components into an uncorrelated vector [27]. Specifically, it converts the image into a principal component vector, and it reconstructs and compresses the image using only the principal components of the first and second (p/2) dimensions. As it evaluates the correlation of the input data statistically, it is an optimal way to measure a state in between known maximum, minimum, and intermediate data.

Now let us describe the KL conversion we used on the SPR curve data. For the conversion explanation, we used SPR measurement data on an osmium mediator-coated electrode when 0.5 V was applied after 100 s of 0-V application using the potentiostat (Figure 5A). To obtain the standard value of the SPR image data, we calculated the average of the SPR curves of the completely oxidized state (0.5 V applied) and completely reduced state (0 V applied) as the reference states. Then, we took the difference from the raw SPR image data as SPR image data deviation (Figure 5A). Next, we performed principal component decomposition on the SPR image data deviation using the following formula.
SPR data deviation = U × D × Q.(1)

Here, U is a matrix of an orthonormal basis, D is a diagonal matrix, and Q is a coefficient matrix. Figure 5B show the diagonal matrix obtained from the SPR image data deviation. It can be seen that the feature quantity is well aggregated in the first component. Finally, we obtained the matrix of KL-converted SPR curve data with the squared error by multiplying the transpose of U by the SPR data deviation. As confirmed in D, only the first component was used as the KL-converted SPR data because the feature quantity was aggregated in the first component. Finally, the value of each state change of the electron mediator was normalized. Figure 5C show the result of converting the SPR curve data before normalization. It can be seen that the KL-converted SPR data before normalization correlates well with the change in the charge on the sensor surface. The result of converting the SPR measurement when the potential was swept like a cyclic voltammetry measurement (Figure 3A) is shown in Figure 6. Between the sweeps from the reduction peak potential to the oxidation peak potential, the KL-converted SPR data decreased in correlation with the current value. It is considered that this result clearly indicates the transition of the state transition reaction of the osmium mediator described in Section 2.1. Furthermore, by combining the SPR curve changes into a single value using KL conversion, it was possible to clearly determine that the applied voltage range in which the osmium state transition reaction largely occurs is 0.2 V to 0.25 V. Based on this information, a protocol for setting the applied voltage to 0.2 V at the time of sample introduction was determined. 

In addition, we collected SPR measurement data on NaCl solutions with different concentrations on gold thin film and compared the KL-converted values with the ordinarily measured SPR angles. The results were highly correlated (Appendix A), which shows that KL-converted SPR data are useful even in a non-high-noise environment.

### 3.3. H_2_O_2_ Detection Using EC-SPR and KL-Converted SPR Data

To validate the idea of using KL-converted SPR data, we tried using KL-converted SPR data to detect H_2_O_2_ quantitatively. H_2_O_2_ is an intermediate molecule that accepts the charge to the electron mediator through the enzymatic reaction of HRP and can be detected by modifying just the electron mediator polymer on the electrode shown in Figure 1. The activity of the target oxidase varies greatly depending on the fixation conditions and solvent conditions, so it can be used for quantitative detection of H_2_O_2_, which has a small number of cascades and a high reaction efficiency.

When measurements of H_2_O_2_ detection at different concentrations were repeatedly performed on the same chip, it was observed that the state of the electron mediator on the electrode changed with each measurement. Figure 7 show data subjected to KL conversion without normalization (almost equivalent to PCA) to measurement data in which measurement under the same conditions (0.5 V Application, 0 V Application) is repeatedly performed. When the number of measurements N = 7, the coefficient of variation was 15% in the 0.5 V applied measurement data, and 8% in the 0 V applied measurement data. This is because repeated measurements on the same chip cause degradation of the mediator on the electrode surface, and PCA also aggregates even small changes, resulting in large differences between measurements. In KL conversion without normalization, the degradation of the mediator and the measurement error is incorporated in the SPR measurement data. For this problem, the concept of KL conversion we have adopted is effective. In other words, in order to calibrate the state of the mediator of each measurement in which variation occurs, a protocol in which a complete oxidation state and a complete reduction state are performed before each measurement, and calibration (standardization) is performed with the obtained data. Thus, since measurement errors can be eliminated, stable measurements that are not easily affected by disturbances are expected. In the electrochemical SPR measurement, the maximum state change corresponds to the complete oxidation state of the mediator, and the minimum state change corresponds to the complete reduction state of the mediator. Since these conditions can be easily set by applying voltage by a potentiostat, the measurement protocol can be adjusted simply by customizing it according to the acquisition of data necessary for analysis.

Figure 8 show results when samples with different H_2_O_2_ concentrations fill the microchannel. All measured SPR data were normalized by the value of the completely oxidized state (0.5 V) and the value of the completely reduced state (0 V) value (Figure 8A). These results indicate that the rate of increase in the KL-converted SPR data increases with the H_2_O_2_ concentration. In SPR measurement, it is known that the maximum rate of change in the measured SPR angle correlates with the concentration of analyte in the sample solution, namely the amount of chemical reaction occurring in the SPR measurement area [1,8]. Therefore, it is considered that the KL-converted SPR data calculated from the SPR curve can be calculated by the same method. Figure 8B plot the maximum rate of increase of the KL-converted SPR data against H_2_O_2_ concentration. The plot shows a linear correlation with the H_2_O_2_ concentration, indicating that the SPR data obtained in this way can be used to determine molecular concentrations in an appropriate measurement environment and that the detection limit of H_2_O_2_ is 0.7 μM. The detection limit is determined as the concentration that changes more than the standard measurement error. We also performed a current measurement, a standard electrochemical measurement, at the same time as the SPR measurement and examined the correlation between the results obtained with KL-converted SPR data and the current measurement (Appendix A). The KL-converted SPR data and current data were linearly correlated. Thus, KL conversion of SPR data proved to be an electrochemically appropriate technique.

### 3.4. Glutamate Detection by Using EC-SPR and KL-Converted SPR Data

As the charge correlation of the KL-converted SPR data and correlation with the electrochemical measurement was confirmed, we performed a quantitative measurement on a small target molecule, glutamate. First, we examined various methods of immobilizing the glutamate oxidase on the electron mediator polymer, i.e., physical adsorption, moisturization by polysaccharide mixing, and inclusion using a polyion complex (PIC) amperometry measurement. As a result, the inclusion method using PIC showed the most efficient charge transfer (Appendix A). Next, we prepared glutamate oxidase, including PIC, on the electron mediator polymer and performed SPR measurements on different concentrations of glutamate sample solution. Figure 9 show the results. The time course of the KL-converted SPR data in Figure 9A varied depending on the concentration of the target molecule, similar to the detection of H_2_O_2_ described above. A calibration curve for glutamate was then prepared using the rate of increase as an index value (Figure 9B). The curve shows a linear correlation between the KL-converted SPR data and glutamate concentration, and the limiting concentration of detection was 5 μM. The reproducibility of the measurements was high, and the coefficient of variation for each concentration was less than 3% (Appendix A). This measurement system is thus considered to be practical because it covers the blood concentration range of 10–50 μM [28]. Moreover, it can easily treat different target molecules simply by changing the oxidase on the electron mediator; as long as the oxidizing enzyme of the target molecule can be prepared, it can be quantitatively measured with this system.

## 4. Conclusions

We found that performing KL-conversion data processing on SPR curve data is useful for EC-SPR measurements and standard SPR measurements even when there is uneven light absorption above the gold sensor surface. In particular, we found that the data processing and EC-SPR can quantitatively measure small molecules; in the case of glutamate, the detection limit was 5 μM. Moreover, we can detect multiple molecule-sized targets at one time by combining the standard SPR and EC-SPR using a 1-D SPR instrument and patterned electrodes in the microchannel. This sensor system reduces the physical burden on users and has the potential for on-site use.

## Figures and Tables

**Figure 1 biosensors-12-00615-f001:**
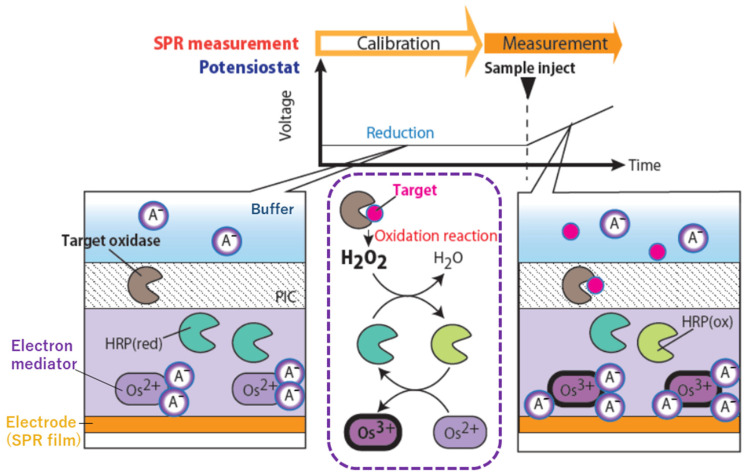
Mechanism of EC-SPR measurement. A- is a charge compensation molecule in a buffer consisting of chlorine ions and phosphate ions. The illustration enclosed in the purple dotted line is the molecular electronic transition reaction cascade from target capture to electron mediator oxidization. Reduced electron mediators attract two charge compensation molecules per molecule, but when oxidases capture the target molecule, the charge of the electron mediator changes, attracting three charge compensation molecules per molecule. Since SPR can detect the molecular concentration on the electrode surface, it can detect the difference in the charge compensation molecular weight attracted by the electron mediator.

**Figure 2 biosensors-12-00615-f002:**
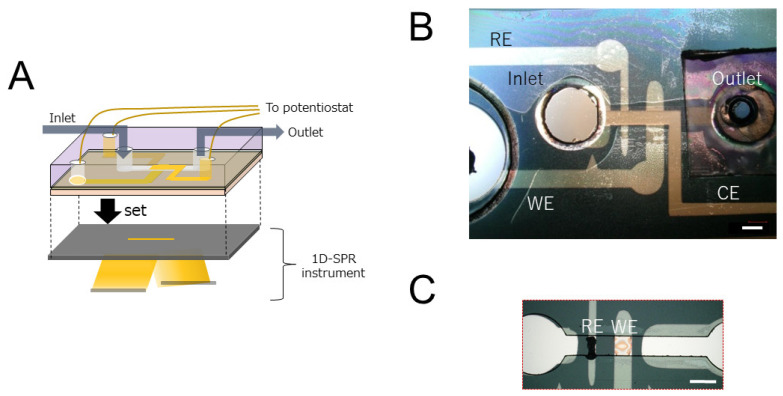
EC-SPR measurement chip layout. (**A**) Chip layout of EC-SPR measurement. (**B**) Photograph of electrodes after the glass substrate and acrylic substrate were bonded. RE: Reference electrode, WE: Working electrode, CE: Counter electrode. (**C**) Photograph of reference electrode and working electrode on glass substrate. Each scale bars are 1 mm.

**Figure 3 biosensors-12-00615-f003:**
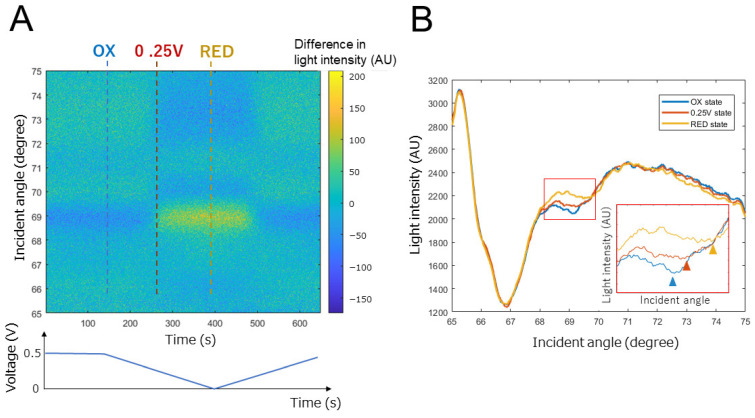
SPR measurements on electrodes subjected to potential sweep. (**A**) Time-course of SPR curve. Light intensity was standardized by zscore. Ox: oxidase state of osmium, 0 V: state with a charge of 0 V, RED: redox state of osmium. (**B**) Changes in SPR curve due to changes in the state of osmium. The inset is an enlarged view of the SPR curve between incident angles of 68 and 70 degrees. The triangles are the secondary dips in each SPR curve.

**Figure 4 biosensors-12-00615-f004:**
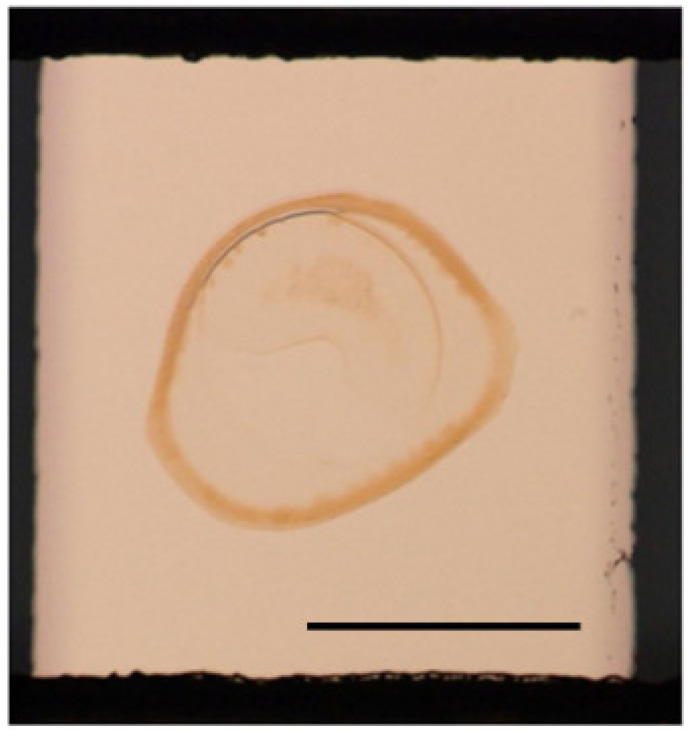
Photograph of electron mediator film applied on the electrode. Scale bar is 0.5 mm.

**Figure 5 biosensors-12-00615-f005:**
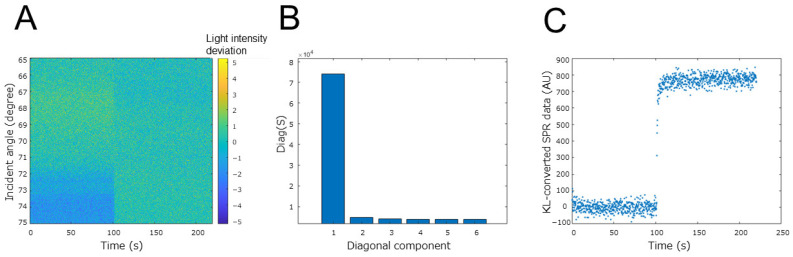
Feature quantity analysis obtained by KL conversion of image data of SPR curve. (**A**) SPR data deviation obtained from the completely oxidized and completely reduced states of the SPR curves. (**B**) Diagonal matrix obtained by KL conversion. (**C**) KL-converted SPR data.

**Figure 6 biosensors-12-00615-f006:**
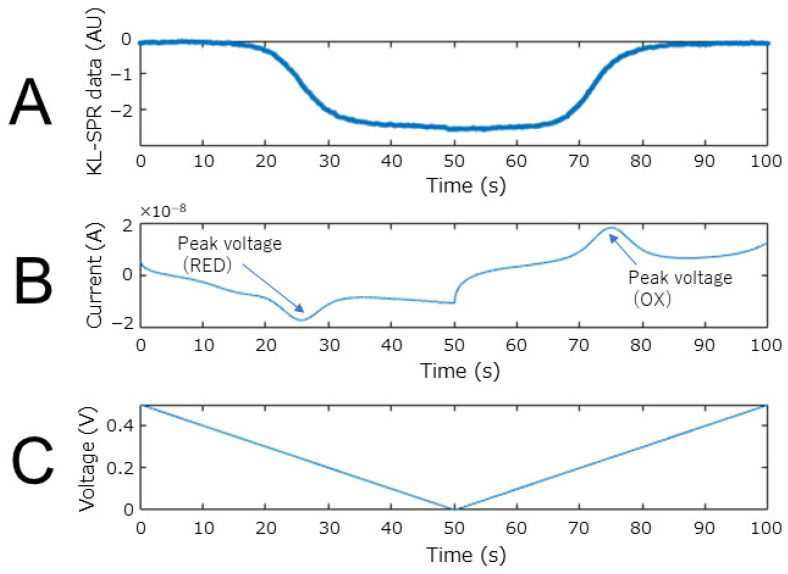
Change in KL-converted SPR curve data when sweeping the potential. (**A**) KL-converted SPR curve data. KL-converted SPR data value was normalized using the 0.5 V value and 0.25 V value. (**B**) Current change detected by potentiostat. (**C**) Electronic sweep.

**Figure 7 biosensors-12-00615-f007:**
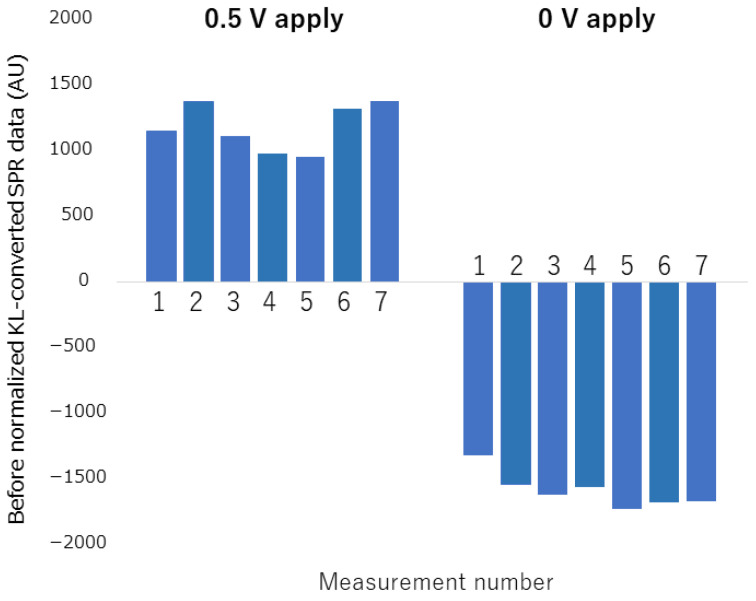
KL-converted SPR measurement data before normalization when repeated measurement is performed under the same voltage application condition.

**Figure 8 biosensors-12-00615-f008:**
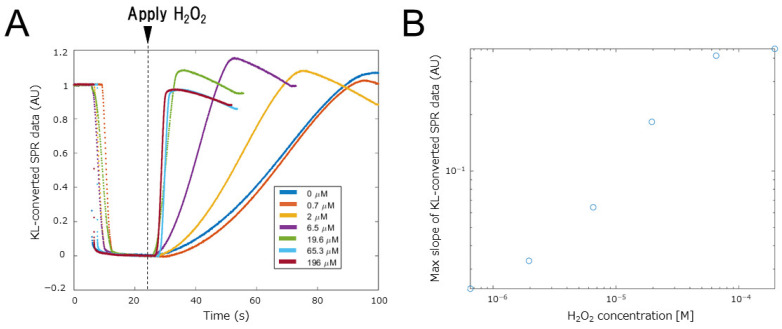
Variation in KL-converted SPR data with H_2_O_2_ concentration. (**A**) Raw KL-converted SPR data showing transition with H_2_O_2_ concentration. (**B**) Calibration curve for H_2_O_2_ detection using the maximum slope of KL-converted SPR data.

**Figure 9 biosensors-12-00615-f009:**
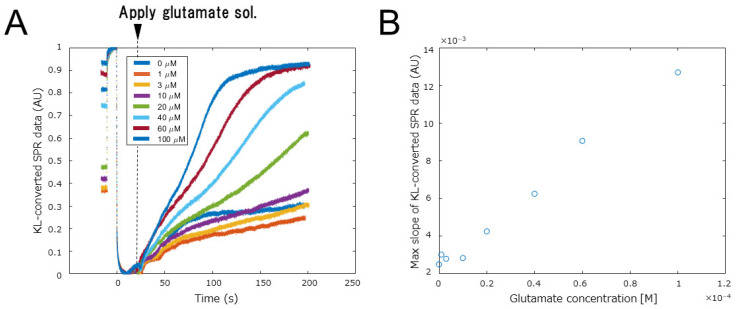
Variation in KL-converted SPR data with glutamate concentration. (**A**) Raw KL-converted SPR data versus glutamate concentration. (**B**) Calibration curve for glutamate detection using the maximum slope of KL-converted SPR data.

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
