# Peer review of "Data Processing of SPR Curve Data to Maximize the Extraction of Changes in Electrochemical SPR Measurements"

_biosensors, 2022, doi:10.3390/bios12080615_

Round 1
Reviewer 1 Report
1) I am not a specialist in the electrochemistry that is why it is not quite clear for me what is depicted in the figure 1, where is a buffer, mediator, electrode, SPR film, etc, and what is depicted by double arrows.
2) Also it is not clear why in the figure 3A there is almost no change in SPR data when voltage starts to reduce from 0.5V and when it bounces from 0, but there is an abrupt change at voltage 0.25V.
3) Further, it is not clear how the time scale in the figure 5A corresponds to the time scale in the figure 3A. Why there is a leap at 100 s?
4) Finally, it is not clear how the detection limits (0.1 uM for H2O2 and 5 uM for glutamate) was determined. Besides, it seems strange that difference between maximum slopes for curves 19.6 uM and 65.3 in the figure 7B looks so big, when corresponding curves in the figure 7A are very close to each other as well as 65.3 and 196 uM.
Reviewer 2 Report
In their work, Inoue et al. report on the development of a data-processing method applied to a hybrid electrochemical-SPR platform and prove the effectiveness of their approach by providing the extracted calibration curves for H2O2 and glutamate detection, with claimed sensing performances that meet the range of interest of the chosen analytes. My main concern regarding this manuscript is the utilization of the Karhunen-Loève conversion itself. It basically coincides with PCA, which is an already employed technique for biosensing data analysis [https://doi.org/10.3390/bios9010008, https://doi.org/10.1016/j.sna.2020.112323, https://doi.org/10.3390/bios10080100, https://doi.org/10.1088/1742-6596/2049/1/012048]. Therefore, the novelty of this paper appears to be insufficient and, personally, I do not recommend this paper for publication in Biosensors. Together with this, I have few more points that are listed afterwards:
1. The introduction is too short and lacks of relevant, non-self-cited literature. It is obviously okay to insert proper self-references but, overall, the bibliography of this paper is poor.
2. The acronym HRP is not explicated.
3. Generally, the figures are poorly readable and the font size in the legends should be increased.
4. Some experimental methods are not properly illustrated (e.g., the detailed procedures used for glutamate oxidase immobilization on the mediator polymer are missing).
5. The English language and style must be improved since it results inattentive and unclear in more than a case. Here are few examples:
LL 180-181: “The observation position showing the largest change in the SPR curve in the range where the mediator was applied was used for analysis.” Do the authors mean that they performed the experiments on the coffee-ring trace? If so, how could the SPR effect be exploited since the analyte would be well far from the gold surface?
LL 202-204: The expressions “the real part of the reflected light” and “the imaginary part of the reflected light” make no sense.
LL 224-226: “Regarding the problem of the dispersion of the feature-quantity displacements in the SPR curve, we consider that data processing aggregates displacements in various parameters to make one large displacement.” What does this sentence mean?
6. The authors use the expression “highly correlated” referring to Fig. S2, but no relevant quantities are provided.
7. Line 298: Glutamine -> Glutamate.
8. Regarding the biodetection part, no information is provided on the number of replicates over which the experiments were conducted, there are no error bars nor trace of the relevant parameters to evaluate the quality of the performed fits (I am referring both to the slope of KL-converted data and to that used for LOD quantification).
Reviewer 3 Report
Peer review report on “Data Processing of SPR Curve Data to Maximize Extraction of Changes in Electrochemical SPR Measurements”
Comments on authors:
1. Overview and general recommendation
In this manuscript, a data-processing method called KL-conversion was used on SPR curve data for EC-SPR measurement and standard SPR measurement. According to the authors, this data processing method is useful to solve the problem of SPR curve data readout error caused by light absorption during EC-SPR measurement and quantitatively detect small molecules.
On the one hand, I found the paper to be overall well written and much of it to be well described. I felt confident that the authors performed careful and thorough experimental processing. And I found the proposed measuring and data-processing method interesting and promising for SPR application. On the other hand, however, I found the description of some important parts inadequate. Also, some writing and language issues in the article need to be revised.
2. Comments
2.1 I suggest that the authors give the rationale for choosing KL as the SPR data processing method, including but not limited to why the KL method was chosen over other methods and what are the positive effects of choosing the KL method for SPR data processing.
2.2 A comparative approach can be used to compare the KL method with other possible data processing methods, reflecting the advantages of the data processing method proposed in this paper, rather than just comparing it with the original SPR data.
2.3 The key part of this article is to perform KL conversion on SPR data and verify its feasibility, which is the H2O2 detection using EC-SPR and KL-converted SPR data and Glutamine detection by using EC-SPR and KL-converted SPR data subsections (The subsection number here is incorrectly labeled). However, While reading this section I was confused as to how this approach addresses the questions posed by the authors and how this approach reflects the correlation with the original SPR data. For example, Page 8 Line 282: These results indicate that the rate of increase in the KL-converted SPR data increases as the H2O2 concentration increases (figure 7A). I think as a newly proposed method the authors should explain in more detail what the KL converted SPR data represents and how it relates to the concentration of the substance to be measured.
2.4 Page 10, Figure 8A: The colors of different concentrations are not clearly labeled, and it is impossible to distinguish the concentration of each color represented by the substance to be measured.
2.5 There are some formatting errors in the manuscript, such as Page 3, Line 117, Page 4, Line 156, Page 5, Line 169, Page 8 Line 272, which all have first-line character indentation problems. The author should double-check.
2.6 Figure S3: The parameter R2 needs to be explained. Then what data do the red lines and blue dots represent?
Round 2
Reviewer 2 Report
The authors have conducted an extensive revision of the paper, provided acceptable point-by-point response to all the comments and addressed the main issues. Most importantly, they have clarified the aim of the presented research. My only suggestion is to decrease the font size of the letters in the figures and increase that of the legends, the axes units and axes names, but this can be managed during the proofing process.
Overall, despite having been quite strict in my previous report, I can now recommend this paper for publication in Biosensors.